# Antiepileptic Drugs and Their Dual Mechanism of Action on Carbonic Anhydrase

**DOI:** 10.3390/jcm11092614

**Published:** 2022-05-06

**Authors:** Calin Magheru, Sorina Magheru, Marcela Coltau, Anica Hoza, Corina Moldovan, Liliana Sachelarie, Irina Gradinaru, Loredana Liliana Hurjui, Felicia Marc, Dorina Maria Farcas

**Affiliations:** 1Department of Medical Disciplines, Faculty of Medicine and Pharmacy, University of Oradea, 410073 Oradea, Romania; magherus@yahoo.com (S.M.); anica_hoza@yahoo.com (A.H.); cmold2003@yahoo.com (C.M.); feliciamarc.dr@gmail.com (F.M.); dmfarcas@yahoo.com (D.M.F.); 2City Hospital Prof. Dr. Ioan Puscas Simleu Silvaniei, 455300 Simleu Silvaniei, Romania; marcelacoltau@yahoo.com; 3Department of Biophysics, Apollonia University, 700511 Iasi, Romania; 4Department of Medical Disciplines, Faculty of Medicine and Pharmacy, “Grigore T. Popa” University of Medicine and Pharmacy, 700115 Iasi, Romania

**Keywords:** antiepileptic drugs, epilepsy, carbonic anhydrase II

## Abstract

(1) Background: The benefit of using inhibitors of carbonic anhydrase (CA), such as acetazolamide, in the treatment of epilepsy has previously been described. (2) Methods: In this paper, the effect on CA of the most well-known antiepileptic drugs was studied in vitro and in vivo. The effects, after chronic treatment, of carbamazepine, phenytoin, valproate, primidone, clonazepam, and ethosuximide were studied in vitro on purified CA, isozyme I (CA I) and CA, and isozyme II (CA II) activity and in vivo on epileptic erythrocyte CA I and CA II activity. (3) Results: In vitro results showed that all antiepileptic drugs reduced purified CA II activity according to dose–response relationships and slightly inhibited CA I activity. In vivo results showed that the chronic administration of antiseizure drugs induced a progressive reduction in erythrocyte CA II activity in all the groups studied. This study shows that CA II inhibition can be induced both in vitro and in vivo by major antiepileptic agents as it might be one of the effective mechanisms of these anticonvulsant drugs. (4) Conclusions: The decrease in CA II activity in epileptic patients after antiseizure treatment suggests the involvement of CA II in the pathogenesis of epilepsy.

## 1. Introduction

Epileptic seizures often cause transient impairment of consciousness, leaving the individual at risk of bodily harm and often interfering with education and employment [1,2]. Therapy is symptomatic in that the available drugs inhibit seizures, but neither effective prophylaxis nor a cure is available. Compliance with medication is a major problem because of the need for long-term therapy together with the unwanted effects of many drugs [2].

The mechanism of action of antiseizure drugs falls into three major categories [3,4]. Drugs that are effective against the most common forms of epileptic, partial, and secondarily-generalized clonic seizures appear to work by one of two mechanisms [5]. One is bylimiting the sustained repetitive firing of a neuron, an effect mediated by promoting the inactivated state of voltage-activated Na^+^ channels [6,7]. A second mechanism appears to involve enhanced gamma-aminobutyric acid (GABA)-mediated synaptic inhibition, an effect that is mediated by an action presynaptically for some drugs and postsynaptically for others [6,8]. Drugs that are effective against a less common form of epileptic seizure limit the activation of a particular voltage-activated Ca^2+^ channel known as the T current [6,9]. Phenobarbital was the first synthetic organic agent that was recognized as having antiseizure activity [10]. The chemical structures of most of the drugs introduced before 1965 were closely related to phenobarbital. These include the hydantoins [11], the oxazolidinediones [12], and the succinimides [13]. The agents introduced after 1965 include benzodiazepines [14], an iminostilbene (carbamazepine) [15], a branched-chain carboxylic acid (valproic acid) [16], and a cyclic analog of GABA (gabapentin) [8].

Carbonic anhydrase (CA) is a zinc-enzyme that catalyzes one of the simplest reactions in an organism—between CO_2_ and water (CO_2_ + H_2_O <-> H+ + HCO_3_^−^)—that has a main role in the acid-base balance [17]. It has been known for over fifty years that CA occurs in the mammalian brain [18], where it probably has a number of physiological roles, e.g., in fluid and ion compartmentation [19], in the formation of cerebrospinal fluid [17], in seizure activity [20], and in the respiratory response to carbon dioxide [21]. Carbonic anhydrase has been implicated in the propagation of seizures and the control of edema in the central nervous system [22,23].

The benefit of using inhibitors of CA, such as acetazolamide, in the treatment of epilepsy has also been described [24,25,26]. The anticonvulsant activity of acetazolamide is induced by direct inhibition of cerebral CA [26]. It has been noticed that the tonic component of convulsions produced by maximal electroshock is released by acetazolamide, with higher resistance for voltage seizures. Acetazolamide induces the regulation of nervous influx transmission to cortical and subcortical levels [27].

The similarities between the effects of CO_2_ and acetazolamide as anticonvulsants have suggested that the role of CA in relationship to seizures is mediated through the CO_2_-buffering system [18]. The increase in CO_2_ production during seizures in the presence of CA activity does not exert an inhibitory feedback effect on neuronal metabolism due to its rapid hydration to bicarbonate. In the absence of CA activity, CO_2_ builds up and impairs the ability of neurons to carry out oxidative metabolism at an optimal rate. Starting from these data, this paper presents a study of the relationship between carbonic anhydrase activity, the antiseizure effect of antiepileptic drugs, and the implication of enzymes in themechanisms of action of these drugs.

## 2. Materials and Methods

### 2.1. Study Design

In vitro: The effect of carbamazepine, phenytoin, phenobarbital, valproate, primidone, clonazepam, and ethosuximide on CA I and CA II purified from human erythrocyte (SIGMA Diesenhofen, Diesenhofen, Germany) was studied. All antiseizure drugs studied here were used as pure substances (SIGMA Diesenhofen, Germany)using, as a standardized solvent, dimethyl sulfoxide(DMSO) (SIGMA Diesenhofen, Germany).Assessments were performed according to dose–response relationships at concentrations of between 10^−8^ and 10^−4^ M.

In vivo: This study was conducted between April 2019 and April 2020 at the Center for Research and Medical Assistance, Simleu Silvaniei, Romania, in accordance with The Declaration of Helsinki, as modified by the 21st World Medical Assembly. A written consent form was obtained from each patient. The research was carried out with the agreement of the Ethics Commission of the Center for Research and Medical Assistance (Simleu Silvaniei, Salaj, Romania), no. 327/24.03.2018.

A group of 168 epileptic patient volunteers aged between 18 and 65, weighing between 51 and 94 kg, and of either sex was selected for this study. All subjects were community-dwelling and in good general health apart from their epilepsy. Subjects were screened before participation with a history and a physical examination, a complete blood count and routine chemistries, urinalysis, and an electrocardiogram. Patients were excluded from participation if they exceeded 135% of their ideal body weight, had a history of other diseases, were taking any medications, had orthostatic hypotension, had evidence from screening tests of an underlying illness, or significant laboratory or electrocardiogram abnormalities. After this screening, 120 patients with epilepsy, aged between 18 and 54, with a bodyweight of 54–87 kg, and of either sex were selected.

All patients in this study had at least 6 partial seizures monthly despite the use of multiple antiepileptic drugs. These patients were able to record their seizures on calendars, which were provided by the physician. A standard EEG was performed on all patients, and they took no medication for seven days before the study. The patients were divided into 6 groups of 20 patients each, according to their type of seizure, and they received an oral treatment with appropriate antiepileptic drugs for 15 days, as follows:-Group 1: patients with partial seizures without any impairment of consciousness who took carbamazepine 400 mg/day; electroencephalography had not revealed a seizure focus.-Group 2: patients with partial complex seizures treated with phenytoin600 mg/day; positive tracings were found in 90% of these patients by recording EEGs during sleep and by using special electrodes positioned close to the temporal lobe structure.-Group 3: primary generalized tonic-clonic seizures treated with valproate250 mg/day; EEGs showed lateral synchronous spikes and wave discharges.-Group 4: secondarily generalized epilepsy treated with primidone 1500 mg/day; EEGs showed the same modifications as in primary generalized tonic-clonic seizures, but the clinical manifestations of a focal onset were variable in anatomy, modality, and degree.-Group 5: primary myoclonic seizures treated with clonazepam10 mg/day; EEGs showed multifocal spikes, polyspikes, and slow waves.-Group 6: unclassifiable seizures, with an adequate history, treated with ethosuximide 1500 mg/day; the routine EEG was abnormal.

In all groups, erythrocyte CA I and CA II activity was measured before and after treatment. The reduction in seizure severity was also estimated, considering postictal alertness, length of seizures, injuries, and the severity of the ictal state. The reduction in seizure frequency was calculated as the percent change in seizure rates during the last 15 days compared with the baseline before this study. Clinically significant seizure reduction was defined as a >50% reduction in seizures (responders).

To determine erythrocyte CA activity in humans, 1 mL of venous blood was taken 2 h after drug administration, and these samples were immediately washed in 9 mL NaCl solution (0.15 M) and centrifuged for 10 min at 3000 rot/min. After centrifugation, the supernatant was removed and the hemolysis of red cells was achieved by suspension in 9 mL of bidistilled water. A 1/10 dilution from the hemolysate was used in the enzymatic assay of CA to measure red cell CA activity.

Differentiation of carbonic anhydrase I from carbonic anhydrase II activity in erythrocytes was performed by testing with nicotinates [28], which relies on the selective inhibition of carbonic anhydrase I activity induced by methyl nicotinate at a concentration of 10^−4^ M. Total CA activity was assayed and noted as Total CA. Methyl nicotinate at a concentration of 5 × 10^−4^ M was associated with the same sample. At this level of concentration, CA I activity was completely inhibited. Afterwards, CA activity in the sample was measured again. This represented the value of CA II activity. From the Total CA activity, the value of CA II activity was subtracted, and the value of CA I activity was obtained.

### 2.2. Materials

Purified CA I and CA II, HEPES, p-nitrophenol, carbamazepine, phenytoin, valproate, primidone, clonazepam, and ethosuximide were purchased from SIGMA (Deisenhofen, Germany). The antiepileptic drugs used were: Tegretol (Ciba, Basel, Switzerland), Phenhydan (Desitin, Hamburg, Germany), Depakine (Sanofi, Paris, France), Mysoline (ICI-Zeneca, London, UK), Rivotril (Hoffmann-La Roche, Basel, Switzerland), and Pentinimid (Gerot, Vienna, Austria).

### 2.3. Technique of CA Assay

The activity of CA was assessed using the stopped-flow method [29]. This method consists of measuring the enzymatic activity of CO_2_ hydration, and it relies on the colorimetric method of changing pH. The time in which the pH of the reagent mixture decreased from its initial value of 7.5 to its final value of 6.5 was measured. Follow-up of the reaction was achieved spectrophotometrically at a wavelength of 400 nm using a rapid kinetic spectrophotometer, HI-TECH SF-51MX (UK), equipped with a mixing unit and a system of two syringes that supplied the reagents. The signal transmitted by a photomultiplier from the mixing chamber was received and visualized using a computer equipped with a mathematical coprocessor and a kinetic software package, RKBIN IS1.

The reagents used were: p-nitrophenol, as a color indicator, at a concentration of 0.2 mM, a pH of 7.5, and a temperature of 20–25 °C; HEPES buffer at a concentration of 20 mM, a pH of 7.5, and a temperature of 20–25 °C; stock solutions of purified CA I and CA II, each at a concentration of 3.44 × 10^−6^ M, a pH of 7.5, and a temperature of 20–25 °C; CO_2_ solution at a concentration of 15 mM (as a substrate), which was obtained by bubbling CO_2_ in bidistilled water to saturation; and Na_2_SO_4_ at a concentration of 0.1 M, which was used to keep up a constant ionic strength.

The activity of carbonic anhydrase was obtained by the formula:A = T_0_ − T [enzyme units/mL](1)
where T_0_ represents the uncatalyzed reaction time, and T represents the catalyzed reaction time (in the presence of CA I or CA II).

In the dioxide carbon (CO_2_) hydration reaction catalyzed by carbonic anhydrase, one enzyme unit caused the pH to drop from 7.5 to 6.5 per minute at 25 °C.

### 2.4. Statistics

To determine whether CA I and CA II activity were affected by antiepileptic drugs, a repeated measures ANOVA was performed. Comparisons between treatments were made using the Neuman–Keuls multiple comparison test. A paired *t*-test was used to compare initial values with the steady-state for a given treatment. Probabilities of *p* < 0.05 were considered significant.

## 3. Results

In vitro: All the antiepileptic drugs studied here significantly inhibited CA II in a dose-dependent manner. The effect set in at 10^−8^ M, and the peak was at 10^−4^ M, as presented in Table 1. The inhibition of purified CA I was highly reduced as compared to that of CA II.

In vivo:

A thirty-day treatment with antiepileptic drugs reduced the activity of erythrocyte CA II and seizure frequency as follows:-Group 1: Carbamazepine reduced the activity of CA II from 1.82 ± 0.08 to 0.86 ± 0.02 EU/mL (*p* < 0.001) (Figure 1). Overall, 16 of 20 patients reported a 72% seizure reduction. Four patients reported a reduction in seizure frequency between 25 and 49%.-Group 2: Phenytoin reduced the activity of CA II from 1.72 ± 0.06 to 0.93 ± 0.03 EU/mL (*p* < 0.05)(Figure 1). Overall, 18 of 20 patients reported an 81% seizure reduction. Two patients reported a reduction in seizure frequency of up to 50%.-Group 3: Valproate reduced the activity of CA II from 1.89 ± 0.07 to 1.10 ± 0.02 EU/mL (*p* < 0.001) (Figure 1). Overall, 14 of 20 patients were responders who experienced a >65% reduction in seizure frequency.-Group 4: Primidone reduced the activity of CA II from 1.71 ± 0.06 to 0.82 ± 0.03 EU/mL (*p* < 0.001) (Figure 1). Overall, 18 of 20 patients reported a >50% reduction in seizure frequency.-Group 5: Clonazepam reduced the activity of CA II from 1.59 ± 0.04 to 1.18 ± 0.02 EU/mL (*p* < 0.05) (Figure 1). The reduction in seizure frequency was >60% in 18 of 20 patients.-Group 6: Ethosuximide reduced the activity of CA II from 1.62 ± 0.05 to 1.22 ± 0.02 EU/mL (*p* < 0.05) (Figure 1). The reduction in seizure frequency was >50% in 16 of 20 patients.

The activity of CA I was not significantly changed in any of the groups presented above (Figure 2).

## 4. Discussion

The in vitro results show that all the antiepileptic drugs currently used in therapy and that belong to different pharmaceutical classes reduce purified CA II activity according to the dose–response relationship. The effect set in at 10^−8^ M and reached a peak at 10^−4^ M, where the maximum effect of inhibition was present with carbamazepine (64.2%), and low inhibition was achieved with phenobarbital (31.6%). These in vitro data also show that the major antiseizure drugs inhibit CA II by a direct mechanism of action. Our in vitro results have revealed that antiepileptic drugs slightly inhibit CA I activity as compared to that of CA II. The major effect was present at a 10^−4^ M concentration for carbamazepine, where the inhibition was 21%.

In vivo chronic administration of antiseizure drugs induced a progressive reduction in erythrocyte CA II activity in all the groups studied.

Inhibition of CA II after the antiepileptic treatment depended on the type of therapy used. Among the antiseizure agents studied in vivo, carbamazepine was the most potent inhibitor of human erythrocyte CA II activity. The inhibition percentage was 52.75% for this drug, and concomitantly patients reported a 72% reduction in seizure frequency.

Erythrocyte CA I activity in epileptic patients wasnot significantly modified after chronic treatment with certain antiepileptic drugs. The isozyme activity presented a slow decrease (8–10%) as compared to the initial value.

Moreover, this study revealed the effectiveness of four antiepileptic drugs—carbamazepine, phenytoin, valproate, and primidone—for the treatment of partial or secondarily-generalized clonic seizures that was associated witha strong reduction in CA II activity. The side effects of these drugs were limited to fatigue, nystagmus, depression, and ataxia. The results of this study also showed that myoclonic or unclassifiable seizures were best treated with clonazepam and ethosuximide and were associated with a moderate reduction in erythrocyte CA-II activity. These drugs produced minor side effects, such as drowsiness, ataxia, and decreased muscle tone.

Regarding the mechanism of action of the antiepileptic drugs studied here, it is known that carbamazepine, phenytoin, and valproate act by depolarization of spinal cord or cortical neurons, and this appears to be mediated by slowing the rate of recovery of voltage-activated Na^+^ channels from inactivation [24,30,31]. In addition, valproate also stimulates the activity of the GABA synthetic enzyme, glutamic acid decarboxylase, andinhibits GABA degradative enzymes [31]. The mechanism by which barbiturates and deoxy barbiturates work involve the potentiation of synaptic inhibition through an action on the GABA_A_ receptor [32], while ethosuximide reduces low-threshold Ca^2+^ currents (T currents) in thalamic neurons [13].

Concerning the correlation between carbonic anhydrase and epilepsy, the studies performed by Puscas and collab. Ref. [27] showed that patients with epilepsy had increased red cell CA activity as compared with controls. Moreover, the serum of epileptic patients activated purified CA, an effect antagonized by proglumid and cimetidine (a specific gastrin antagonist and histamine antagonist, respectively). Their results suggest the implication of CA in the pathogenesis and therapy of epilepsy.

Correlating those findings with the results of this study, data herein suggest that anti-epileptic drugs might have a dual mechanism of action: the first consisting of their action on Na^+^ channels, Ca^2+^ current, or the GABA system; and the second mechanism, proposed in this study, which shows that antiepileptic drugs inhibit carbonic anhydrase with consecutive pH increases. These increases in pH might be an additional factor involved in the intracellular mechanism of action of antiepileptic drugs. Hypoxia and cerebral edema (due to various causes, such as brain trauma, stroke, brain neoplasm, alcohol, and drug use) are known to increase neuronal excitability in the brain, causing changes in affected neurons, including changes in electrical exchanges and ion channels lowering the local pH and increasing CO_2_. Increasing the concentration of AC II produced by antiepileptic medication by increasing the local pH helps to reduce the number and intensity of epileptic seizures. AC is a glial enzyme, and glucose, neuronal loss, and brain atrophy are associated with epilepsy [32,33]. This dual mechanism of action would bring new data regarding the anticonvulsant effects of carbamazepine, phenytoin, and the most frequently used antiseizure drugs, the effects of which might also be parallel and dependent on the inhibition of carbonic anhydrase II, which is a glial enzyme. It was demonstrated that an increase in CA activity induced an increase in glial cell activity, which reduced the extent of the spread, and the intensity, of seizures by enhancing the ability of the glial cells to regulate acid-base homeostasis in the neurons and the interstitial fluid of the brain [33].

## 5. Conclusions

This study led to the conclusion that inhibition induced by antiepileptic drugs of CA II in brain tissue might be associated with a reduction in seizures. Our results suggest that CA II inhibition induced by the major antiepileptic agents both in vitro and in vivo might be one of the mechanisms of action of these anticonvulsant drugs.

This study may be a precursor to other research on certain drugs that increase the activity of carbonic anhydrase to counteract the side effects of anticonvulsant medications and make them more tolerable.

## Figures and Tables

**Figure 1 jcm-11-02614-f001:**
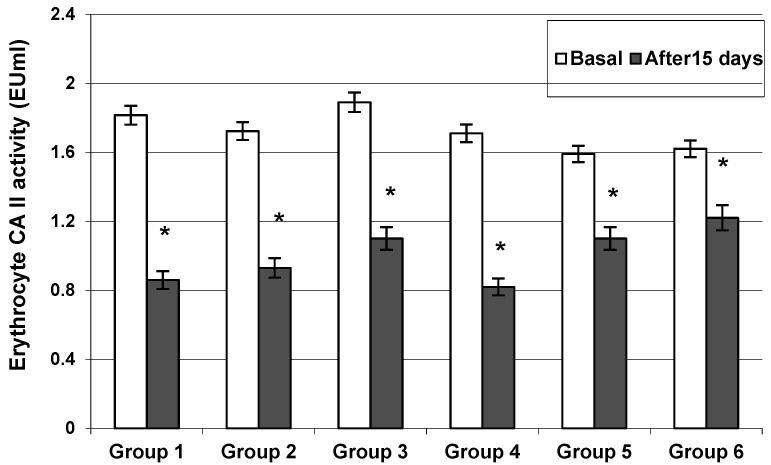
The inhibition of erythrocyte Carbonic Anhydrase II activity after 15 days of treatment with carbamazepine (Group 1), phenytoin (Group 2), valproate (Group 3), primidone (Group 4), clonazepam (Group 5), and ethosuximide (Group 6). Values are means ± SEM; *n* = 20 patients; * significant differences (*p* < 0.05) after comparing values before treatment (paired *t*-test).

**Figure 2 jcm-11-02614-f002:**
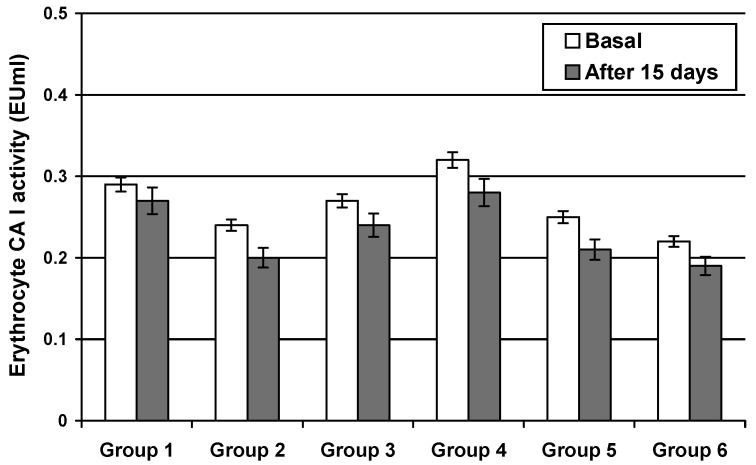
Erythrocyte Carbonic Anhydrase I activity after 15 days of treatment with carbamazepine (Group 1), phenytoin (Group 2), valproate (Group 3), primidone (Group 4), clonazepam (Group 5), and ethosuximide (Group 6). The isozyme activity presented a slow decrease as compared to the initial values. Values are means ± SEM; *n* = 20 patients.

**Table 1 jcm-11-02614-t001:** The effect of antiepileptic drugs on carbonic anhydrase isozymes.

Substance	Conc. (mol/L)	Purified CA I Basal Activity = 0.450 ± 0.01 EU/mL	Purified CA II Basal Activity = 1.000 ± 0.01 EU/mL
	**10** ^ **−8** ^	0.406 ± 0.01	0.708 ± 0.03 *
**Carbamazepine**	**10** ^ **−6** ^	0.394 ± 0.02	0.521 ± 0.02 *
	**10** ^ **−4** ^	0.347 ± 0.01 *	0.358 ± 0.01 *
	**10** ^ **−8** ^	0.417 ± 0.01	0.812 ± 0.01 *
**Phenytoin**	**10** ^ **−6** ^	0.402 ± 0.02	0.628 ± 0.03 *
	**10** ^ **−4** ^	0.386 ± 0.02 *	0.486 ± 0.02 *
	**10** ^ **−8** ^	0.443 ± 0.01	0.894 ± 0.01
**Phenobarbital**	**10** ^ **−6** ^	0.421 ± 0.03	0.755 ± 0.02 *
	**10** ^ **−4** ^	0.398 ± 0.02	0.684 ± 0.02 *
	**10** ^ **−8** ^	0.435 ± 0.02	0.763 ± 0.01 *
**Valproate**	**10** ^ **−6** ^	0.402 ± 0.02	0.572 ± 0.03 *
	**10** ^ **−4** ^	0.379 ± 0.03 *	0.435 ± 0.02 *
	**10** ^ **−8** ^	0.426 ± 0.01	0.815 ± 0.01 *
**Primidone**	**10** ^ **−6** ^	0.411 ± 0.03	0.624 ± 0.01 *
	**10** ^ **−4** ^	0.383 ± 0.04 *	0.419 ± 0.03 *
	**10** ^ **−8** ^	0.441 ± 0.02	0.872 ± 0.01 *
**Clonazepam**	**10** ^ **−6** ^	0.421 ± 0.01	0.689 ± 0.02 *
	**10** ^ **−4** ^	0.408 ± 0.02	0.603 ± 0.04 *
	**10** ^ **−8** ^	0.419 ± 0.03	0.793 ± 0.02 *
**Ethosuximide**	**10** ^ **−6** ^	0.391 ± 0.01	0.597 ± 0.01 *
	**10** ^ **−4** ^	0.358 ± 0.02 *	0.419 ± 0.03 *

Data are presented as means ± S.E.; *n* = 5 assessments; * significant differences (*p* < 0.05) after comparing basal activity for each isozyme. Conc.: Concentration; CA: Carbonic anhydrase

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
