# Peer review of "Antiepileptic Drugs and Their Dual Mechanism of Action on Carbonic Anhydrase"

_jcm, 2022, doi:10.3390/jcm11092614_

Round 1
Reviewer 1 Report
This point of view is new and interesting but there are some points to improve.
Major Points
#1 All of these experiments used antiepileptic drugs as experimental groups whose effects and mechanisms on CA are not clear. Therefore, it is not possible to assess whether the effects on CA are pharmacological or due to some nonspecific effect. Experiments with positive control drugs such as acetazolamide and negative control drugs that have been proven to have no inhibitory effect on CA are needed.
#2 What are the possible mechanisms by which each antiepileptic drug inhibits CA2?
#3 Are the concentrations of antiepileptic drugs of 10^-8 to 10^-4 in in vitro experiments reasonable compared to the concentrations actually used in the CNS in humans?
Author Response
The authors acknowledge the useful observations and suggestions of the reviewer’s as concerns the manuscript entitled: Antiepileptic drugs and their dual mechanism of action on carbonic anhydrase by
Calin Magheru * , Sorina Magheru , Marcela Coltau , Anica Hoza , Corina Moldovan , Liliana Sachelarie * , Gradinaru Irina * , Loredana Liliana Hurjui * , Felicia Marc , Dorina Maria Farcas
According to the reviewer’s recommendations, all the suggestions were taken into account, as follows:
Major Points
All of these experiments used antiepileptic drugs as experimental groups whose effects and mechanisms on CA are not clear. Therefore, it is not possible to assess whether the effects on CA are pharmacological or due to some nonspecific effect. Experiments with positive control drugs such as acetazolamide and negative control drugs that have been proven to have no inhibitory effect on CA are needed.
What are the possible mechanisms by which each antiepileptic drug inhibits CA2?
R:
All the antiepileptic drugs studied had decreased the activity of eritrocytar AC II the decrease being depending on the used therapy (by increasing pH). The differences may appear due to different mechanisms of action of the antiepileptic drugs: carbamazepine, phenytoin, and valproate produce modification of the Na channels, in addition, valproate stimulates the activity of the GABA synthetic enzyme, glutamic acid decarboxylase and inhibits GABA degradative enzymes. Etosuccimide reduces lo-threshold Ca2+ currents in thalamic neurons.
Are the concentrations of antiepileptic drugs of 10^-8 to 10^-4 in in vitro experiments reasonable compared to the concentrations actually used in the CNS in humans?
R: Yes, especially the concentration of 10^-4M
Thank you very much for review reports and for the extremely useful observations and suggestions!
Kind regards,
Prof.dr. Liliana Sachelarie
Reviewer 2 Report
- Manuscript should be English spell checked eg: In Abstract check spelling of “indused” line 26.
- In recent studies, there are seizures/prolonged seizures (status epilepticus) reported due to hypoxia and edema. Please cite these papers PMID: 33063874; PMID: 10541231 and discuss the relationship between CA, edema, hypoxia, and role of CO2. Discussion of these will definitely expand readability of this paper.
- Discussion paragraph 2 is just one line, Please restructure your discussion
Author Response
The authors acknowledge the useful observations and suggestions of the reviewer’s as concerns the manuscript entitled: Antiepileptic drugs and their dual mechanism of action on carbonic anhydrase by
Calin Magheru * , Sorina Magheru , Marcela Coltau , Anica Hoza , Corina Moldovan , Liliana Sachelarie * , Gradinaru Irina * , Loredana Liliana Hurjui * , Felicia Marc , Dorina Maria Farcas
According to the reviewer’s recommendations, all the suggestions were taken into account, as follows:
- Manuscript should be English spell checked eg: In Abstract check spelling of “indused” line 26.
R: Induced
We checked the article from the point of view of languages errors and grammatical errors.
- In recent studies, there are seizures/prolonged seizures (status epilepticus) reported due to hypoxia and edema. Please cite these papers PMID: 33063874; PMID: 10541231 and discuss the relationship between CA, edema, hypoxia, and role of CO2. Discussion of these will definitely expand readability of this paper.
R: Hypoxia and cerebral edema (due to various causes: brain trauma, stroke, brain neoplasm, alcohol and drug use) are known to increase neuronal excitability in the brain, causing changes in affected neurons, including changes in electrical exchanges and ion channels lowering the local pH and increasing Co2. Increasing the concentration of AC II produced by antiepileptic medication by increasing the local pH), helps reduce the number and intensity of epileptic seizures. AC is a glial enzyme, and glucose, neuronal loss, and brain atrophy are associated with epilepsy. [34,35].
- Discussion paragraph 2 is just one line, Please restructure your discussion
“In vitro” discussion is in the first line and the next lines is talking about the “in vivo” study
Thank you very much for review reports and for the extremely useful observations and suggestions!
Kind regards,
Prof.dr. Liliana Sachelarie
Round 2
Reviewer 1 Report
Authors adequately responded to all the concerns.